# Exploring the Resource Value of Transvaal Red Milk Wood (*Mimusops*
*zeyheri*) for Food Security and Sustainability: An Appraisal of Existing Evidence

**DOI:** 10.3390/plants9111486

**Published:** 2020-11-04

**Authors:** Abiodun Olusola Omotayo, Enioluwa Jonathan Ijatuyi, Adebayo Isaiah Ogunniyi, Adeyemi Oladapo Aremu

**Affiliations:** 1Food Security and Safety Niche Area, Faculty of Natural and Agricultural Sciences, North-West University, Private Bag X2046, Mmabatho 2745, North West Province, South Africa; 25301284@nwu.ac.za; 2Department of Agricultural Economics and Extension, North-West University, Private Bag X2046, Mmabatho 2790, North West Province, South Africa; 24818879@student.g.nwu.ac.za; 3International Food Policy Research Institute (IFPRI), Abuja 901101, Nigeria; a.ogunniyi@cgiar.org; 4Indigenous Knowledge Systems Centre, Faculty of Natural and Agricultural Sciences, North-West University, Private Bag X2046, Mmabatho 2745, North West Province, South Africa

**Keywords:** ethnobotanical uses, functional food, hidden hunger, micronutrients, indigenous fruit, phytochemicals, sapotaceae, sustainability, traditional food

## Abstract

Many African countries are endowed with rich biodiversity with enormous nutritional and economic prospects, but the majority of these resources are not fully harnessed. Exploring these neglected resources, for example, the Transvaal red milkwood tree [(*Mimusops zeyheri*) Sond. family: Sapotaceae] is of paramount importance for food and nutritional security as well as economic prosperity. This review provides a critical appraisal of the nutritional and health benefits as well as the economic potential of *Mimusops zeyheri*. The plant is known for its diverse uses among rural communities. In folk medicine, the decoction from the bark and leaves of *Mimusops zeyheri* are used for treating wounds and ulcers, while the root is used as an infusion taken to treat candidiasis and other health issues. The nutritional profile of the fruit tree is similar to popular exotic fruits and richer in vitamin C when compared to guava and orange. *Mimusops zeyheri* is a rich source of vitamins, protein, and fatty acids. Based on the rich chemical pool, especially in the fruit and seeds, it has the potential to provide an accessible, readily available, and affordable enriched functional food with valuable health benefits. However, the successful exploration of *Mimusops*
*zeyheri* for food security and sustainability requires multidisciplinary research. This will help achieve the envisaged food-nutrition security and poverty alleviation potential of the plant, especially among local communities.

## 1. Introduction

Combating malnutrition and poverty remains key in the face of the increasing global population as more than 820 million individuals (translating into one out of every nine people) in the world faced hunger in 2018. This highlights the immense challenge of achieving the United Nations Sustainable Development Goal (UN SDG Agenda 2030) number 2, which targets zero hunger by 2030 [1]. In Africa, an estimated 20% of the population is chronically undernourished (in terms of dietary and energy supply), making the continent a key region with a high prevalence of hunger [2]. Currently, Africa is lagging in the drive to meet the goals of ending hunger and ensuring access to food for all [3]. In addition, micro- and macro-nutrient deficiencies, as well as ill-health, remain major concerns, especially in rural communities [4]. This explains the importance of understanding the constraints faced by rural households in achieving food security and good health as well as economic liberation and development [5,6]. As a result, exploring opportunities on how undervalued and neglected fruit trees could contribute toward achieving UN SDG Agenda 2030, particularly Goal 2: zero hunger remains pertinent [7,8,9,10]. 

In southern Africa, the problem of food insecurity and ill-health remains prevalent. Specifically, inaccessibility and low financial capacity/purchasing power to afford the popular commercially available fruits. This problem remains common in rural communities where the diets mainly consist of monotonous foods, mainly starchy, and a few legume staples [11]. Thus, there should be more focus on promoting indigenous fruit trees with food and nutrition potential as these species thrive well in their natural environment [12]. The southern African region is endowed with high biodiversity, which has the potential to contribute and strengthen the food system to meet the nutritional needs of the population. The biodiversity potential can be explored and harnessed to improve food security in the region and Africa at large [13,14]. Furthermore, the diversity of diets via exploring indigenous fruit trees including *Mimusops zeyheri* (Transvaal red milkwood tree) could serve as alternative sources of nutrition, with health and socio-economic benefits. 

Currently, the limited information on the nutritional and health value of *Mimusops zeyheri* remains fragmented. The availability and critical appraisal of existing knowledge on the fruit tree have the potential to contribute toward mitigating malnutrition and food insecurity in the southern African region and beyond. Therefore, this review explored the nutritional and health benefits as well as the economic potential of the underutilized *Mimusops zeyheri* as a fruit tree for potential sustainability. The following questions were explored in this review:What are the nutritional and ethnomedicinal uses of *Mimusops zeyheri*?What are the challenges associated with the potential commercialization and sustainability of *Mimusops zeyheri*?How can the economic potential of *Mimusops zeyheri* be fully explored?

## 2. Materials and Methods

Following the approach described by Moher et al. [15], the Preferred Reporting Items for Systematic Reviews and Meta-Analyses (PRISMA) was used in the selection of articles in this review. We explored various online databases, dissertations/thesis, and reports using the library of the North-West University. A thorough search was conducted primarily by using search engines such as Scopus, Web of Science (WOS), and PubMeb, some other quick checks were conducted on Google Scholar, Web of Science, and Science Direct using the following terms and phrases: “*Mimusops zeyheri*”, “Transvaal red milkwood”, “nutritional value composition of Transvaal red milkwood”, “ethnomedicinal importance of the *Mimusops zeyheri*”, “uses of *Mimusops zeyheri*” and “description of Transvaal red milkwood”. For the current review, the search was limited to southern Africa with a literature search period (years) from 1968 to 2020 (August). 

In this review, studies that fit the inclusion criteria were downloaded to assess the content. Four emergent themes were identified and categorized as (i) description and ecology of *Mimusops zeyheri*; (ii) uses of *Mimusops zeyheri*; (iii) challenges; and (iv) prospects of *Mimusops zeyheri* (Table 1). In cases where articles covered more than one of these themes, categories were assigned based on the theme that was most comprehensively addressed in the research questions or findings of the respective article. The process of article selection, summarization, and coding was performed using Microsoft Excel and Word, resulting in an expected minimum variability. 

A total of 186 scientific publications were retrieved that focused on *Mimusops zeyheri* and related indigenous fruits. Finally, an estimated 33% (59) of the scientific publications were eligible and included in the current review (Figure 1). 

## 3. Taxonomy, Morphology, and Distribution of *Mimusops zeyheri*

*Mimusops zeyheri* is a perennial fruit tree and a member of the family Sapotaceae and the common names include Transvaal red milkwood (English), Moepel (Afrikaans), Mmupudu (Northern Sotho), Umpushane (Zulu), Mubululu (Venda) and Mgamba kapu (Swati) [16]. It is a medium-sized tree ranging from 15–25 m high and the bark is grey to dark brown [16]. The leaves are glossy and arranged spirally, stipules are absent, and the petiole is 0.5–3.5 cm long [16]. The fruit is yellow-orange, 4.5 cm long at maturity, with 1–2 seeds that are shiny, but pale brown with a small circular basal scar. 

The tree occurs in several southern African countries including South Africa, the Kingdom of Eswatini (Swaziland), Botswana, Lesotho, Namibia, Mozambique, and Zimbabwe [17,18]. As shown in Figure 2, its distribution extends to other parts of tropical Africa. *Mimusops zeyheri* occurs in open, dry, and bushveld woodland [19,20]. It is suited for cultivation in low to medium-altitude areas of summer rainfall where frost is minimal or absent and can withstand various soil and climate conditions [21]. In larger gardens and open spaces, *Mimusops zeyheri* is easily cultivated and best used as a reliable shade and used as a shelter for birds and other animals [22]. Furthermore, *Mimusops zeyheri* can tolerate freezing without damage and it requires little maintenance (six or more hours of daily sunlight and a minimal amount of water to survive). 

## 4. Diverse Uses of *Mimusops zeyheri*

The continuous increase in the human population remains a key cause for concern in terms of meeting the daily food, nutritional, and medicinal requirements, especially in developing countries. Therefore, documenting the benefits and potential of *Mimusops zeyheri* is important from nutrition and health as well as ethnomedicinal perspectives. This is also relevant for the perception and socio-economic benefits of the potential consumers for this undervalued fruit tree. Some of these uses are discussed in the following sections.

### 4.1. Ethno-Medicinal Applications

In traditional medicine, *Mimusops zeyheri* extracts are used for different purposes. For instance, Lapeña et al. [23] observed that the bark of *Mimusops zeyheri* is used to treat wounds and sores by the Zulus of South Africa. It is also a local remedy for sexually transmitted infections (STIs), especially gonorrhea [24]. In the Kingdom of Eswatini (Swaziland), a root infusion is taken to treat candidiasis and a bark decoction is a traditional remedy for treating wounds and ulcers [25]. The ground seeds of the fruit tree are used as a teeth whitening agent [26]. *Mimusops zeyheri* is considered an effective remedy for treating inflammation, bleeding gums, tuberculosis, and diverse sexually transmitted diseases [25]. The bark and root of *Mimusops zeyheri* are used to treat different forms of wounds and ulcers. The root of *Mimusops zeyheri* is used as an infusion for treating candidiasis, tuberculosis, weight loss, womb problems, and STIs [27,28].

### 4.2. Nutritional Content

Evidence of the diverse nutritional pool existing in different parts of *Mimusops zeyheri* has been demonstrated by different researchers (Table 2). According to Wilson and Downs [29], the fruit of *Mimusops zeyheri* contains high concentrations of sucrose, glucose, and fructose. The orange color of the fruit is an indication of its high *beta*-carotene content. The leaves of *Mimusops zeyheri* contain 10 elements such as nitrogen, phosphorous, potassium, calcium, and magnesium [28]. The quantity for these 10 elements was within the permissible limits, thereby suggesting its safety [25]. Analysis of the fruit revealed varying levels of carbohydrates and ash content [30,31,32]. Also, the seeds of *Mimusops zeyheri* are considered as a dietary energy supplement and oil source [33,34]. 

Analysis of the seeds revealed the presence of different mineral elements including calcium and phosphorus, organic matter, crude protein, and ash content [33]. Based on the proximate analysis, dry matter, organic matter, and ash content constituted 91.1%, 88.3%, and 2.8% of the seed mass, respectively. Chivandi et al. [33] identified 17 amino acids in the seeds of *Mimusops zeyheri*, which accounted for an estimated 97% of the crude protein content (9.3%). Glutamic acid, which was the major amino acid constituted approximately 13.8% of the crude protein. Furthermore, oleic and palmitic acids were the only lipids that were detected. Neutral detergent fiber and acid detergent fiber constituted 33.2% and 15.3%, respectively [32,33].

*Mimusops zeyheri* contains varying concentrations of the different compounds (Table 2). Particularly, the plant is a rich source of vitamins (A = retinol, C = ascorbic acid and E), amino acids, and fatty acids [33,34]. In addition, the medium fiber content in the *Mimusops zeyheri* seeds could be useful in providing the necessary bulk for the facilitation of normal gastrointestinal motility in the food and feeds of humans and animals, respectively [33]. Based on the presence of vitamin E, *Mimusops zeyheri* seeds could potentially be used as a dietary ingredient to increase the systemic antioxidant pool, thus protecting the body against potential oxidative damage.

The lipid in *Mimusops zeyheri* seeds is similar to that found in soybean (*Glycine max*). The lipid content of 21.3% is comparable to that of soybean, which has between 15 and 25% lipid content [32]. However, the oil content of *Mimusops zeyheri* seeds is lower than that of the cottonseed (*Gossipium hirstum*), which has 35–40% oil content [33]. Furthermore, the oleic acid content of *Mimusops zeyheri* seed oil (85% of lipid yield (84.59) is high when compared to the 70–78% oleic acid in Marula tree (*Sclerocarya birrea*) kernel oil and 63% oleic acid in red sour plum (*Ximenia caffra*) kernel oil [35]. Interestingly, these two (2) aforementioned indigenous fruit trees flourish in a similar environment as *Mimusops zeyheri* in southern Africa. The vitamin C content in *Mimusops zeyheri* fruit range from 50–80 mg/g fresh fruit and this content is relatively higher when compared to guava, which is an exotic fruit with 20 mg/g vitamin C [21,36].

### 4.3. Phytochemicals and Biological Properties of Mimusops zeyheri

*Mimusops zeyheri* is a member of the family Sapotaceae, which are well-known for their wide range of phytochemicals, especially saponins, flavonoids, and polyphenolics [37]. Currently, the biological activities of *Mimusops zeyheri* are relatively scarce, however, studies on members of the genus *Mimusops* have demonstrated antifungal, gastro-protective, and antinociceptive properties [37].

### 4.4. Miscellaneous Uses

Other uses of *Mimusops zeyheri* have been documented in areas where the species exists. The importance of the fruit and pulp of *Mimusops zeyheri* for the production of jelly, alcoholic, and non-alcoholic beverages have been indicated [7]. These products are currently sold in rural and urban open markets, thereby contributing to the economic status of rural households [33,38]. *Mimusops zeyheri* is believed to improve soil fertility and this is a good contribution toward environmental sustainability as this reduces the cost of agricultural production [26,38]. In addition, the latex from the tree is used as a pesticide and the dried pulp can stay long, stored, and for consumption during winter [39]. Furthermore, *Mimusops zeyheri* is a useful multipurpose tree, which could be economically useful for furniture and other carpentry purposes. It is also a good horticultural tree for shade, energy, fencing, and ornamental purposes [25]. 

Based on the study by Chivandi, Davidson, and Erlwanger [35], *Mimusops zeyheri* has enormous economic and environmental benefits for rural communities through the sales of seedlings and/or fruits and combating land degradation. In addition, Ripple et al. [39] identified the seeds of *Mimusops zeyheri* as a potential source of calcium, which could be useful for human and animal nutrition requirements.

## 5. Challenges Associated with Potential Commercialization of *Mimusops zeyheri*

In recent decades, the increasing importance of indigenous fruits as a source of nutrition security has been recognized. However, there are challenges associated with the production of many indigenous fruit trees including *Mimusops zeyheri* [34]. Identifying the challenge(s) is the first step in the attempt to devise potential solutions to overcome these bottlenecks. Some of the challenges currently affecting the potential commercialization of *Mimusops zeyheri* fruit are highlighted below.

### 5.1. Inadequate Information on the Effective Cultivation Protocols and Agronomic Aspects

The propagation and establishment of *Mimusops zeyheri* plantations are important for its economic development. Presently, the cultivation of indigenous fruit trees including *Mimusops zeyheri* remains an unpopular investment when compared to exotic fruits. *Mimusops zeyheri* predominantly exists in the wild, which may be linked to the inadequate knowledge that currently exists on their propagation. These trees usually remain in the vegetative propagation state for a long duration before flowering and fruiting. The longevity in the maturation of the *Mimusops zeyheri* fruit tree will make this unattractive to smallholder farmers, who need a regular income for their livelihood and economic survival. According to Akinnifesi et al. [40,41], the reduction of fruiting time for wild fruit trees will strongly encourage investment and interest from stakeholders. Therefore, investment focused on the elucidation and shortening of the reproductive cycle and development of *Mimusops zeyheri* remains essential.

### 5.2. Inadequate Information on the Biological Efficacy, Therapeutically Value-Added Products with Commercial Prospect

Presently, *Mimusops zeyheri* has several uses in folk medicine [25,28,30], but the majority of these lack scientific evidence based on responses in biological systems. The inadequate evidence-based data to support the biological potential is a major challenge that hinders the commercialization of the plant. The global spread of various diseases such as cardiovascular disease, cancer, and diabetes calls for the exploration of more natural resources including plant-based therapeutic agent(s). However, studies on the biological efficacy of *Mimusops zeyheri* including the ideal preparation and processing methods are currently inadequate. 

### 5.3. Lack of Holistic Research Approach, Inadequate Political Will, and Policy Framework

A major challenge associated with *Mimusops zeyheri* is related to awareness, research, political, and socio-economic inadequacies. There is currently inadequate knowledge of several important aspects such as breeding, trait-selection, storability, and shelf-life of *Mimusops zeyheri* fruit. The limited knowledge on its shelf life, ambient conditions, and longevity during preservation under cold storage negatively affects the commercialization potential for this plant. 

Given that *Mimusops zeyheri* is widely consumed by wild animals due to its sweetness, uncontrolled grazing is another major challenge mitigating its exploration. Other concerns include the high exploitation and the indiscriminate harvesting of *Mimusops zeyheri* as fuel, expansion of settlements and agriculture land, and collection of rootstocks for grafting [25]. The ineffective implementation of existing legislation for the protection and management of many forest trees has contributed to the challenges facing *Mimusops zeyheri*. For instance, Lapeña, Turdieva, Noriega, and Ayad [23] indicated that many decision-makers are reluctant to enact laws that protect wild plants.

According to Baldermann et al. [42], inefficiency in production, storing, and processing of the seeds together with a lack of the baseline knowledge of the nutritional potential negatively affect the acceptance, utilization, and value addition of many wild fruits. The fruit species is known for its slow growth [16], which may be a major deterrent to investment by commercial farmers. Despite the hardy nature of *Mimusops zeyheri*, it serves as a host for invasive fruit flies [43], thereby threatening its commercial prospect.

## 6. Prospects of Unlocking the Potential of *Mimusops zeyheri*

### 6.1. Production of New Products with Diverse Applications

The high tolerance of *Mimusops zeyheri* to drought, salt, and resistant to root-knot nematodes can be employed in agriculture for combating the menace of pests and diseases in plants [21,26]. In addition, the exploration of the seeds for oils can be pursued. Extensive research on *Mimusops zeyheri* may result in the development of a wide range of cosmetic products while the gel can be useful in the pharmaceutical sector for novel drug delivery systems [44,45]. To enhance the potential to develop new products for both humans and animals, researchers need to focus more on the biological, phytochemicals, and nutritional content of *Mimusops zeyheri*. As highlighted by Leakey [46], the successful domestication of indigenous species can enhance its potential in local, regional, and international markets and contribute to the bio-economy.

### 6.2. Devising an Effective Value Chain

To unlock the potential of *Mimusops zeyheri*, an effective value chain highlighting the sequence and critical activities from the production to the marketing stage of the fruit tree and its product is proposed (Figure 3). The development of cultivation, preharvest, and postharvest protocols to enhance the production of indigenous plants for consumption will impact the four pillars of food security, namely access, supply, availability, and utilization, which is essential for a food secured country [47]. The key-players required for the value chain of *Mimusops zeyheri* will include a wide range of stakeholders such as farmers, transporters, wholesalers, and retailers as well as the consumers. Presently, the sales of *Mimusops zeyheri* serve as a marginal portion of livelihood among smallholder farmers who harvest from the wild population [36]. Rural communities with potential for cultivating this fruit tree are plagued with prevailing subsistence farming, perhaps due to poor financial strength or lack of essential farm inputs, which happens in a cyclical order and continually.

### 6.3. Efficient Agro-Processing

As described by Mhazo et al. [48], agro-processing involves activities that transform agricultural commodities into different forms that improve handling, increase shelf-life, and add value to the final product. Efficient agro-processing aligned with the drive for the modernization of agriculture and food production systems in Africa [49], thereby enhancing the income and livelihood of households in rural communities. Therefore, the development of small-scale *Mimusops zeyheri* agro-processing enterprises could initiate the path toward the commercialization of *Mimusops zeyheri* and associated products. This type of approach is known to reduce waste of fresh produce, and increase the participation of rural households in commercial economies [50]. Agro-processing of *Mimusops zeyheri* can contribute to sustainable livelihoods through improved incomes, employment, food availability, nutrition, and social and cultural well-being of the rural populace. Small-scale food processing activities represent a potential source of livelihood for many poor people in developing countries.

The overall potential of agro-processing of *Mimusops zeyheri* is huge as it is capable of: (a)Increasing the value of the output of poor farmers thus, higher returns and improved livelihood;(b)Expanding market opportunities for the farmers;(c)Extending the shelf-life and improving the palatability of the fruit;(d)Increasing diversity of value-added products;(e)Enhancing food security by reducing food losses while increasing availability and accessibility;(f)Overcoming seasonality and perishability constraints; and(g)Empowering women who are often more involved in agro-processing activities than men.

### 6.4. Application of Appropriate Biotechnological Methods for Plant Improvement and Cultivation

From a molecular perspective, *Mimusops zeyheri* is currently under-studied. A starting point could be to collect the range of *Mimusops zeyheri* varieties for development as well as to characterize them as a principal source of primary germplasm. Considering the urgent need for food-nutrition as well as the need for an immediate intervention to increase agro-biodiversity, *Mimusops zeyheri* could form a significant source of accessible plant genetic resource(s). Generally, there is a need to fully elucidate the intra-and inter-genetic relationship in wild plants as well as the identification of unique attributes existing across different geographic populations using relevant biotechnological tools [51,52].

In southern Africa, the genetic diversity of *Mimusops zeyheri* remains anecdotal despite the biodiversity potential of the region. This may be due to the poor documentation of the genetic diversity of indigenous resources in the region and the continent at large [52,53]. There is a need to assess the current genetic diversity for *Mimusops zeyheri* to facilitate its conservation and development. Even though *Mimusops zeyheri* possesses several attributes that could drive rural communities and agricultural development, the lack of adequate information and research has created a disincentive for possible interventions [44]. This also affects the development, investment, and entry into formal market systems.

Furthermore, the genetic variations within *Mimusops zeyheri* accessions can potentially create a problem of heterogeneity. Hence, biotechnological tools together with conventional crop genetics, breeding techniques as well as *de novo* domestication can be employed for the rapid genetic improvement of *Mimusops zeyheri*. Agricultural biosciences and innovation need to be strengthened to ensure food security, enhance nutrition, and improve health in developing countries. Therefore, unlocking the resource value of *Mimusops zeyheri* to meet the needs of potential consumers remains pertinent. The need for cheap, affordable, and accessible fruits presents opportunities for developing countries to capitalize on their biodiversity by exploring the resource value of their indigenous fruit trees including *Mimusops zeyheri* to capture niche markets.

## 7. Concluding Remarks

*Mimusops zeyheri* has been traditionally used by rural peoples in southern Africa. Based on the rich pool of nutrients in the fruit, it remains an affordable, accessible, and nutritionally superior alternative to some relatively expensive “exotic” fruits. Evidence also suggests that *Mimusops zeyheri* has the potential to meet the immediate nutritional requirement as it is edible in the fresh stage upon harvesting. This strongly suggests its ability in providing dietary diversity, especially in resource-limited communities. In addition, the agroforestry benefits of this fruit tree to the smallholder farmers could contribute toward increased farm productivity and reduction of farm inputs such as organic fertilizers, pesticides, and herbicides. Based on preliminary data, the commercial potential of the oil from *Mimusops zeyheri* seeds needs to be further explored. However, several challenges associated with the potential commercialization of the fruit tree currently exist. Some of these include inadequate information on the effective cultivation protocols and agronomic aspects, lack of holistic research approach as well as inadequate political will and policy framework. Prospects of unlocking the potential of *Mimusops zeyheri* fruit tree need to include effective value-chain, effective agro-processing, and application of appropriate biotechnological methods for plant improvement and cultivation. Overall, the successful implementation of an ideal *Mimusops zeyheri* fruit requires multidisciplinary research that covers fields and expertise in plant breeding, micropropagation, and agronomy as well as pharmaceutical, toxicological, and phytochemical aspects. There is also a need for appropriate socio-economic and marketing policies. Finally, sufficient investments by governments, non-governmental organizations, and other relevant stakeholders remain pertinent. 

## Figures and Tables

**Figure 1 plants-09-01486-f001:**
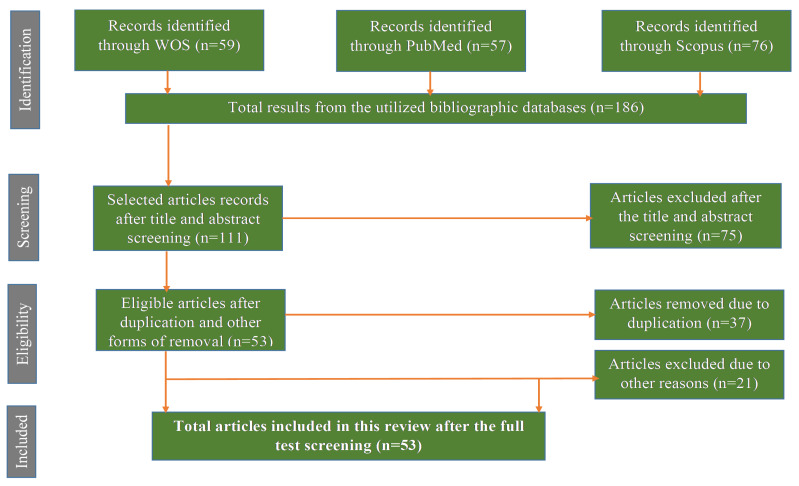
Preferred Reporting Items for Systematic Reviews and Meta-Analyses (PRISMA) flow diagram for the exclusion and inclusion of articles in the current review.

**Figure 2 plants-09-01486-f002:**
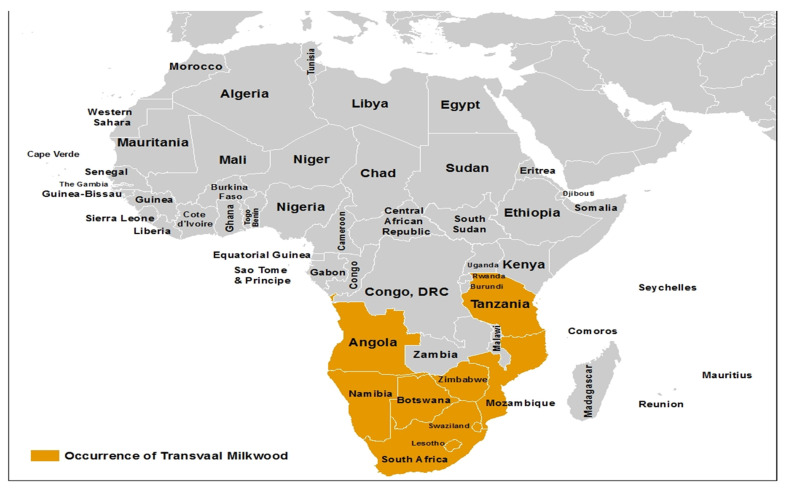
Occurrence and distribution of Transvaal red milkwood (*Mimusops zeyheri*) in Africa.

**Figure 3 plants-09-01486-f003:**
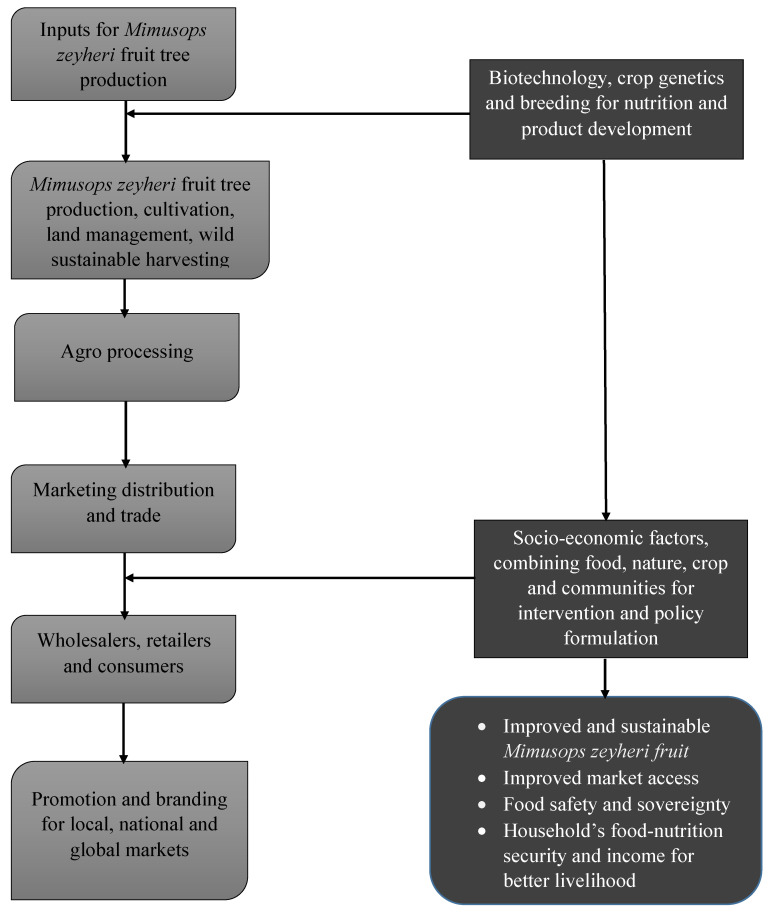
Potential value chain for the undervalued *Mimusops zeyheri.*

**Table 1 plants-09-01486-t001:** Selection criteria for scientific publications used in the current review.

Exclusion Criteria	Explanation
Undervalued African plants	Articles on various edible and non-edible fruit plants.
Undervalued Southern African plants	Articles on various edible fruits
History and horticulture	Articles on archaeological evidence, historic use, domestication, and cultivation of wild edible fruits
Chemical composition	Articles on chemical and nutrient composition, medicinal use of edible fruits
Non-edible uses	Articles describing uses of wild fruits other than food
**Inclusion Criteria**	
Primary study subject is resource value of *Mimusops zeyheri* fruit tree	Studies on the uses, nutritional composition, benefits, and prospects of *Mimusops zeyheri*
Description and ecology	Articles on regional diversity, ecological dynamics, taxonomy, morphology, and distribution of *Mimusops zeyheri*
Diverse uses	Articles documenting the ethnomedicinal, nutritional, biological, phytochemicals, and economic uses of *Mimusops zeyheri*
Challenges	Issues relating to delay in commercialization of the *Mimusops zeyheri*
Prospects	Articles documenting food value chain, trade, markets, and supply chains, policy, and interventions.

**Table 2 plants-09-01486-t002:** Mineral, proximate, and amino acid composition for different parts of *Mimusops zeyheri.*

Plant Part	Nutrient Composition	Composition (%)	Reference
Leaves	Nitrogen	6.33	[25]
	Phosphorous	0.33	
potassium	1.25	
Calcium	0.39	
Magnesium	0.06	
Zinc	0.0029	
Copper	0.0014	
Iron	0.0409	
Aluminum	0.007407	
Manganese	0.005185	
Fruit	Dry matter	91.10	[33]
	Organic matter	83.30	
Protein	9.30	
Ash content	2.80–4.1	
Carbohydrates	2.0	[25]

Seeds	Mineral	Mean value (mg per 100 g)	[33]
	Calcium	587.4	
	Magnesium	102.3	
	Phosphorus	110.37	
	Proximate component	Mean value (g/kg)	[33]
	Dry matter	911	
	Organic matter	883.39	
	Crude protein	93.45	
	Ether extract	256.12	
	Lipid yield	212.5	
	Ash	27.61	
	Fiber fraction (g/kg)	Mean	[33]
	Neutral detergent fiber	332.46	
	Acid detergent fiber	153.41	
	Fatty acid	Percentage (%)	[33]
	16:0 (palmitic acid)	15.25	
	18:1n9 (oleic acid)	84.59	
	Vitamin	μg/g	[33]
	Vitamin E	1.97

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
