# Peer review of "Exploring the Resource Value of Transvaal Red Milk Wood (*Mimusops"

_plants, 2020, doi:10.3390/plants9111486_

Round 1

Reviewer 1 Report

Nice work, which gives an exhaustive review of the food and medicinal use of

Mimusops zeyheri and its possible application for the food security stronghold. Authors have worked through publications of the last 52 years and have proposed research questions to be addressed by future studies. 

Reviewer 2 Report

The paper provides a review of the literature available on an indigenous African fruit tree, which itself is a subject of potential interest.

Remarks of the reviewer:

- The English of the article must be revised, major linguistical corrections are needed (see e.g. "ill health"-L 47, "information which are available"-L 67, L 102, L 160, L 206-207, etc.)

- What is "the goal 2, SDG" (L 54)?

- L166-168: the text and Table 2 are not in line (data referring to seed/fruit are confusing)

- L 173: does the fruit contain retinol or beta-carotene? Is reference 34 appropiate for this data?

- Table 3 contains redundant data (see L 166-168)

- The abstract of the paper refers to phenolic compounds and antioxidant potential of the fruit, however, no mention of these is found in the text. The review should be completed by discussion of phytochemicals other than vitamins.

- The Introduction is excessively long in terms presentation of global food security matters which is not the main topic of the review. 

- Although the Authors refer to the lack of data on the physiological potential of the fruit, no reference is made to any available toxicological data which would be essential for the envisaged valorisation of this food.

Round 2

Reviewer 2 Report

The Authors have substantially improved both the scientific content and the linguistical aspects of their paper. However, still minor corrections are needed regarding the language (e.g. L276, L 291, L 316-317, L 345-346, etc.). Also, the keyword "antioxidants" should be removed.
